# DST-Det: Simple Dynamic Self-Training for Open-vocabulary Object Detection

## Abstract

Open-vocabulary object detection (OVOD) aims to detect the objects *beyond* the set of classes observed during training. This work presents a simple yet effective strategy that leverages the zero-shot classification ability of pre-trained vision-language models (VLM), such as CLIP, to directly discover proposals of possible novel classes. Unlike previous works that ignore novel classes during training and rely solely on the region proposal network (RPN) for novel object detection, our method selectively filters proposals based on specific design criteria. The resulting sets of identified proposals serve as pseudo-labels of potential novel classes during the training phase. This self-training strategy improves the recall and accuracy of novel classes without requiring additional annotations or datasets. We further propose a simple offline pseudo-label generation strategy to refine the object detector. Empirical evaluations on three datasets, including LVIS, V3Det, and COCO, demonstrate significant improvements over the baseline performance without incurring additional parameters or computational costs during inference. In particular, compared with previous F-VLM, our method achieves a 1.7% improvement on the LVIS dataset. We also achieve over 6.5% improvement on the recent challenging V3Det dataset. Our method also boosts the strong baseline by 6.4% on COCO. The code and models will be publicly available.

## 1 Introduction

Object detection is a fundamental task in computer vision, involving localization and recognition of objects within images. Previous detection methods (Ren et al., 2015; He et al., 2017; Lin et al., 2017) are limited to detecting only *predefined categories* learned during the training phase. This limitation results in a considerably smaller vocabulary compared to human cognition. Although directly extending the categories of large datasets would be an ideal solution, it requires an overwhelming amount of manual annotation. Recently, Open-Vocabulary Object Detection (OVOD) (Zareian et al., 2021; Zhou et al., 2022b; Gu et al., 2022; Lin et al., 2023) has emerged as a promising research direction to overcome the constraints of a fixed vocabulary and enable the detection of objects beyond predefined categories.

Typical solutions of OVOD rely on pre-trained VLMs (Radford et al., 2021; Jia et al., 2021). These VLMs have been trained on large-scale image-text pairs and possess *strong* zero-shot classification capability. Prior works (Zareian et al., 2021; Gu et al., 2022; Zhou et al., 2022b) have attempted to leverage VLMs for OVOD by replacing learned classifiers in traditional detectors with text embeddings derived from VLMs. Leveraging VLMs' exceptional zero-shot classification ability enables the detector to assign objects to novel classes based on their semantic similarity to the embedded text representations. Moreover, several recent approaches (Gu et al., 2022; Lin et al., 2023; Wu et al., 2023a) aim to distill knowledge from VLMs by aligning individual region embeddings with visual features extracted from VLMs via diverse distillation loss designs. This alignment process facilitates the transfer of semantic understanding from VLMs to the object detectors. Additionally, some studies (Kuo et al., 2023; Xu et al., 2023; Yu et al., 2023; Wu et al., 2023b) attempt to build open-vocabulary detectors directly upon frozen visual foundation models. Although these methods have demonstrated impressive performance in detecting novel objects, a substantial gap exists between training and testing for novel classes when the vocabulary size (Gupta et al., 2019; Wang et al., 2023a) is larger, since all novel classes are seen as background classes in training. For example, a

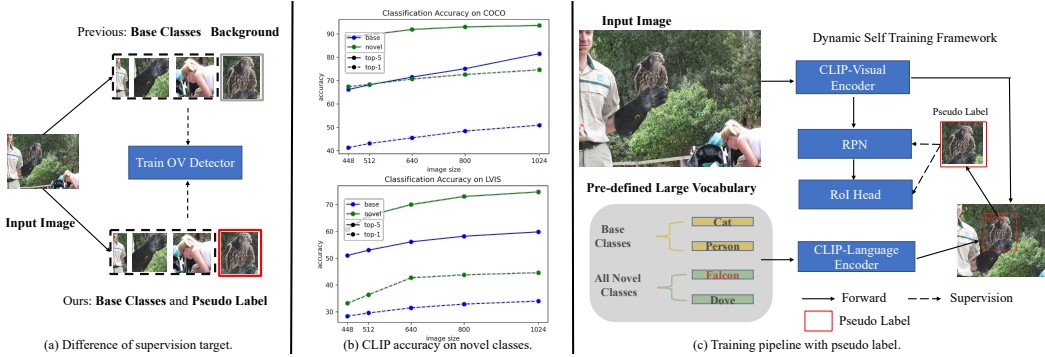

Figure 1: Illustration of our motivation and framework. **(a).** Our DST-Det incorporates novel class labels to supervise the detection head during training. **(b).** Experiments on OV-COCO and OV-LVIS using CLIP with ground truth box for zero-shot classification. We observe high top-1 and top-5 accuracy in classifying novel classes. **(c).** Illustration of our dynamic self-training pipeline with the pseudo labels.

more recent dataset V3Det (Wang et al., 2023a) contains over 10,000 classes. There are over 5,000 classes as novel classes. However, during training, all these classes are treated as background.

In this work, we rethink the training pipeline of OVOD and propose a new dynamic self-training approach by exploiting the novel class knowledge of VLMs. As shown in Fig. 1(a), all previous OVOD methods adopt the same training pipeline by using the base class annotations and treating novel objects as background. During testing, the novel objects are discovered by the region proposal network (RPN). Thus, a conceptual gap exists between the training and testing phases when dealing with novel classes. Our approach aims to bridge this gap by utilizing the outputs of CLIP (Radford et al., 2021) models as pseudo labels during the training phase.

To substantiate our motivation, we first conduct a toy experiment as illustrated in Fig. 1(b), where we calculate the top-1 and top-5 accuracy for both base classes and novel classes on LVIS and COCO datasets by using CLIP's visual features and text embeddings. Specifically, we use the ground-truth boxes to obtain visual features from the feature maps of CLIP and calculate cosine similarity with text embeddings for zero-shot classification. Our results show that the top-5 accuracy for novel classes on LVIS and COCO datasets is already *remarkably* high. This observation inspires us to consider directly using CLIP outputs as pseudo labels during the training phase. As shown in Fig. 1(a, c), we present the DST-Det (dynamic self-training detection) framework, which directly adopts the large vocabulary information provided by CLIP for training.

To reduce the extra computation cost of the CLIP visual model, we let the object detection and the pseudo-labeling generation share the frozen CLIP visual backbone. This decision was supported by recent developments in exploiting frozen foundation models (Kuo et al., 2023; Wu et al., 2023b; Yu et al., 2023). Our DST-Det is based on the two-stage detector Mask R-CNN (He et al., 2017) and incorporates a dynamic pseudo-labeling module that mines novel classes from negative proposals during training. Those proposals with high novel class scores can be considered foreground objects. Adopting simple threshold and scoring operation, we effectively suppress noises in the pseudo labels, such as substantial background contents and useless image crops. These designs *do not* incur any additional learnable parameters for novel class discovery and the process in dynamic since region proposals vary in each iteration. We apply this operation to both the RPN and the Region-of-Interest Head (RoIhead) in the training stage: during RPN training, we force the discovered novel objects to be foreground objects; during RoIHead training, we add the novel class labels directly to the classification target. Moreover, we propose an offline refinement process using the trained detector to generate pseudo labels, boosting performance since the final detector has converged and can directly output more stable proposals.

We show the effectiveness of DST-Det on the OV-LVIS (Gupta et al., 2019), OV-V3Det (Wang et al., 2023a) and OV-COCO (Lin et al., 2014) benchmarks. Our proposed method consistently outperforms existing state-of-the-art methods (Lin et al., 2023; Wu et al., 2023a; Kuo et al., 2023) on LVIS and V3Det without introducing any extra parameters and inference costs. Specifically, DST-Det achieves 34.5% rare mask AP for novel categories on OV-LVIS. With the Faster-RCNN

framework (Ren et al., 2015), our method achieves 13.5% novel classes mAP on V3Det, which boosts previous methods by 6.8 % mAP. Moreover, our method improves considerably on the smaller vocabulary dataset COCO compared to the baseline method. We also provide detailed ablation studies and visual analyses, both of which validate the effectiveness of the DST-Det framework. Moreover, compared with previous pseudo label methods (Gao et al., 2022; Huynh et al., 2022), our method does not use external datasets or extra learnable network components, which makes our approach a plug-in-play method for various datasets.

## 2 RELATED WORK

**Close-Set Object Detection.** Modern object detection methods can be broadly categorized into one-stage and two-stage. One-stage detectors (Redmon et al., 2016; Liu et al., 2015; Tian et al., 2021; Lin et al., 2017; Zhou et al., 2019; 2021; Zhang et al., 2019; Tan et al., 2020b) directly classify and regress bounding boxes using a set of predefined anchors, where anchors can be defined as corners or center points. Recently, several works have adopted query-based approaches (Carion et al., 2020; Zhu et al., 2021; Sun et al., 2021; Liu et al., 2022) to replace the anchor design in previous works. Meanwhile, long-tail object detection aims to address class imbalance issues in object detection. To tackle this challenge, various methods have been proposed, such as data re-sampling (Gupta et al., 2019; Liu et al., 2020; Wu et al., 2020), loss re-weighting (Ren et al., 2020; Tan et al., 2020a; 2021; Zhang et al., 2021; Wang et al., 2021), and decoupled training (Li et al., 2020; Wang et al., 2020). However, all these methods cannot be generalized to novel categories. Our method focuses on the open-vocabulary detection setting.

**Open-Vocabulary Object Detection (OVOD).** This task extends the detector's vocabulary at the inference time, where the detector can recognize objects not encountered during the training. Recently, OVOD (Zareian et al., 2021; Gu et al., 2022; Zhong et al., 2022; Zang et al., 2022; Zhou et al., 2022b; Du et al., 2022; Feng et al., 2022; Wu et al., 2023a; Rasheed et al., 2022; Li* et al., 2022; Lin & Gong, 2023; Gao et al., 2022; Zhao et al., 2022; Ma et al., 2022; Minderer et al., 2023; Arandjelovi'c et al., 2023; Zhang et al., 2023; Kaul et al., 2023a; Cho et al., 2023; Song & Bang, 2023; Shi & Yang, 2023; Bravo et al., 2022; Minderer et al., 2022; Chen et al., 2022; Wang & Li, 2023; Buettner & Kovashka, 2023; Shi et al., 2023) has drawn increasing attention due to the emergence of vision-language models (Radford et al., 2021; Jia et al., 2021). On the one hand, several works (Gu et al., 2022; Wu et al., 2023a; Du et al., 2022; Zang et al., 2022) attempt to improve OVOD performance by letting the object detector learn knowledge from advanced VLMs. For instance, ViLD (Gu et al., 2022) effectively distills the knowledge of CLIP into a Mask R-CNN (He et al., 2017). DetPro (Du et al., 2022) improves the distillation-based method using learnable category prompts, while BARON (Wu et al., 2023a) proposes to lift the distillation from single regions to a bag of regions. On the other hand, F-VLM (Kuo et al., 2023) directly builds a two-stage detector upon frozen VLMs (i.e. CLIP) and trains the detector heads only. It fuses the predicted score and the CLIP score for open-vocabulary recognition. Moreover, several works (Kim et al., 2023a;b) aim for better VLMs pre-training for OVOD. Both types of open-vocabulary object detectors treat novel objects as background during training but target them as foreground objects in testing. This gap hinders the detector from discovering objects of novel categories. We aim to bridge this gap by leveraging stronger vision-language models, enabling improved detection performance for novel classes. Several works (Gao et al., 2022; Huynh et al., 2022) also propose pseudo labels to improve OVD tasks. However, these works require *extra VLM tuning and large caption datasets* for pre-labeling, which makes the pipeline complex. In contrast, our method is simpler and elegant, *without* extra pipelines or parameter tuning.

**Vision-Language Pre-training and Alignment.** Several works (Radford et al., 2021; Jia et al., 2021; Kim et al., 2021; Li et al., 2021; 2022) study the pre-training of VLMs in cases of improving various downstream tasks, including recognition, captioning, and text generation. In particular, contrastive learning has achieved impressively aligned image and text representations by training VLMs on large-scale datasets, as demonstrated by several works (Radford et al., 2021; Jia et al., 2021; Alayrac et al., 2022). For example, the representative work, CLIP (Radford et al., 2021), can perform zero-shot classification on ImageNet (Deng et al., 2009) without fine-tuning or re-training. Inspired by CLIP's generalizability in visual recognition, various attempts have been made to adapt CLIP's knowledge to dense prediction models such as image segmentation (Xu et al., 2023; Rao et al., 2022; Ghiasi et al., 2022; Zhou et al., 2022a) and object detection (Gu et al., 2022; Zang et al., 2022) in the context of open-vocabulary recognition. Meanwhile, several works (Zhou et al., 2022a;

Shi et al., 2022) use CLIP to extract pseudo labels for dense prediction tasks. Our method effectively explores CLIP's ability in the case of OVOD, where CLIP models help discover the object of novel categories to bridge the conceptual gap between training and testing.

## 3 METHOD

### 3.1 PRELIMINARIES

**Problem Setting.** Given an image I, the object detector should output a set of boxes $b_i$ and their corresponding class labels $c_i$. Here, $b_i$ is a vector of length four representing the coordinates of the bounding box around the object, $c_i$ is a scalar indicating the class label assigned to that object. The OVOD detector has a significantly larger vocabulary size for class labels. During the training phase, the OVOD detector can only access the detection labels of base classes $C_B$. But it is required to detect objects belonging to both the base classes $C_B$ and the novel classes $C_N$ at test time. The novel objects are unavailable during the training and are always treated as the background.

**Architecture Overview.** Most previous OVOD methods adopt a two-stage detector. Therefore, we take the Mask R-CNN (He et al., 2017) as an example to demonstrate how it can be adapted to the OVOD task by leveraging the textual information from pre-trained vision-language models. Mask R-CNN consists of two stages: a Region Proposal Network (RPN) and a Region-of-Interest Head (RoIHead). The RPN denoted as $\Phi_{\mathrm{RPN}}$ generates object proposals, and the RoIHead denoted as $\Phi_{\mathrm{RoI}}$ refines the proposals' locations and predicts their corresponding class labels. This process can be formulated as follows:

$$\{r_i\}_{i=1}^M = \Phi_{\mathrm{RoI}} \circ \Phi_{\mathrm{RPN}} \circ \Phi_{\mathrm{Enc}}(\mathrm{I}), \tag{1}$$

$$\{b_i, c_i\}_{i=1}^M = \{\Phi_{\mathrm{box}}(r_i), \Phi_{\mathrm{cls}}(r_i)\}_{i=1}^M, \tag{2}$$

where $\Phi_{\mathrm{Enc}}$ is an image encoder [1] that maps the input image I to a series of multi-scale feature maps. $M$ is the number of proposals generated by $\Phi_{\mathrm{RPN}}$. $\Phi_{\mathrm{RoI}}$ extracts the region embedding $r_i$ from the feature maps given a box proposal. Then the box regression network $\Phi_{\mathrm{box}}$ refines the coordinates of the bounding boxes, and the classification network $\Phi_{\mathrm{cls}}$ predicts the class label of the object within the bounding box. We use $\circ$ to represent the cascade of different components.

The classification head is learnable in closed-vocabulary object detection and maps the region embedding into predefined classes. However, in the open-vocabulary scenario (Zhou et al., 2022b; Gu et al., 2022), the classifier is substituted with text embeddings generated by pre-trained VLMs (i.e., CLIP) and is frozen during the training. The text embedding $t_c$ for the $c$-th object category is generated by sending the category name into a CLIP text encoder using either a single template prompt, "a photo of category," or multiple template prompts. And for a region embedding $r$, its classification score of $c$-th category is calculated as follows:

$$p_c = \frac{\exp(\tau \cdot <r, t_c>)}{\sum_{i=0}^C \exp(\tau \cdot <r, t_i>)} \tag{3}$$

where $< \cdot, \cdot >$ is the cosine similarity, and $\tau$ is a learnable or fixed temperature to re-scale the value.

**Frozen CLIP as Backbone.** To reduce computation cost and strengthen the open-vocabulary ability of the object detector, we use the CLIP image encoder as the detector backbone in $\Phi_{\mathrm{Enc}}$. We keep the parameters of our backbone fixed to preserve the vision-language alignment during training following F-VLM (Kuo et al., 2023). Then, the inference stage of the detector can benefit from both the detection score described in Eq. 3 and the CLIP score obtained by replacing the region embedding in Eq. 3 with the CLIP representation of the corresponding region proposal as depicted in Fig. 2(c). Specifically, we apply the RoIAlign function $\Phi_{\mathrm{RoI}}$ to the top-level feature map of $\Phi_{\mathrm{Enc}}$. Then, the CLIP representations of the region proposals are obtained by pooling the outputs of the RoIAlign via the attention pooling module of the CLIP image encoder. Given a region proposal during the inference, the score of the $c$-th candidate category is obtained by merging the two types of scores via geometric mean:

$$s_c = \begin{cases} p_c^{1-\alpha} \cdot q_c^{\alpha} & \text{if } i \in C_B \\ p_c^{1-\beta} \cdot q_c^{\beta} & \text{if } i \in C_N \end{cases} \tag{4}$$

---

[1]In this paper, we ignore the Feature Pyramid Network (FPN) (Lin et al., 2017) for brevity.

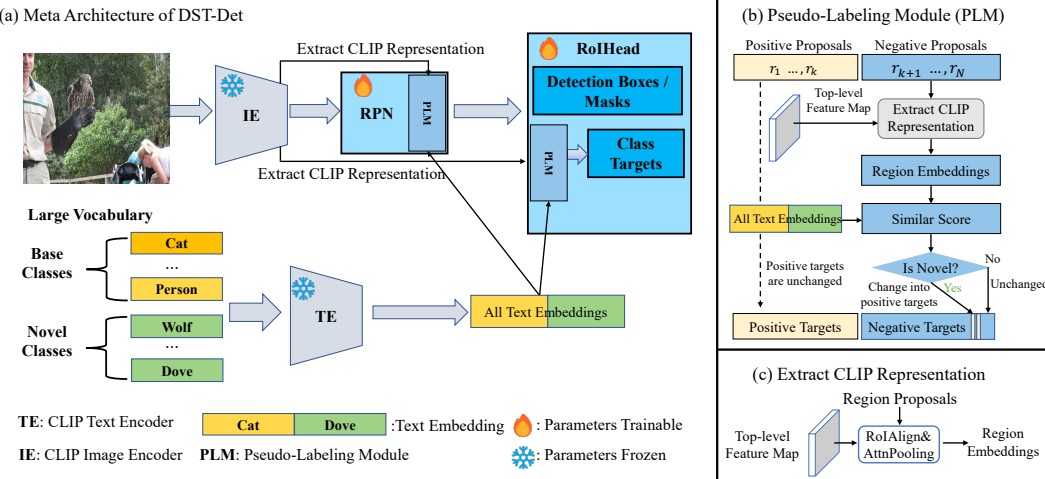

Figure 2: Illustration of DST framework. **(a)** The meta-architecture of DST-Det. **(b)** The proposed pseudo-labeling module (PLM). The PLM is inserted into the two stages of the detector, including the RPN and RoIHead. During training, PLM takes the top-level feature map from the CLIP image encoder and text embedding of object classes as input and generates the pseudo labels for the RPN and the RoIHead. **(c)** The process of extracting CLIP representation for region proposals. The RoIAlign operation is applied to the top-level feature map, the output of which is then pooled by the Attention Pooling layer (AttnPooling) of the CLIP image encoder.

where $q_c$ is the CLIP score, $\alpha, \beta \in [0, 1]$ control the weights of the CLIP scores for base and novel classes, and $C_B$ and $C_N$ represent the base classes and novel classes, respectively.

## 3.2 DST-DET: DYNAMIC SELF-TRAINING FOR OVOD

**Motivation of DST-Det.** Previous works have addressed the problem of open-ended classification in OVOD by utilizing the text embeddings from pre-trained VLMs (e.g., CLIP). However, a conceptual gap still exists between training and testing for novel classes. As shown in Fig. 1, annotations for novel objects are considered background during training. However, during the test phase, they are expected to be detected as foreground and classified into a specific novel class based on their detection score. A new training strategy is needed to bridge the gap between the training and test phases by utilizing CLIP's outputs as pseudo labels for novel classes. In the toy experiments depicted in Fig. 1(b), we treat the ground-truth bounding boxes as region proposals and get the CLIP score for each region proposal, and use the CLIP scores to verify the zero-shot classification ability of CLIP on OV-COCO and OV-LVIS datasets. How to get the CLIP score has been illustrated in Sec. 3.1 and Fig. 2(c). The results indicate that the top-5 accuracy suffices for discovering novel classes, which motivates us to consider using CLIP outputs as pseudo labels during training, given the RPN proposals and a large pre-defined vocabulary.

**Pseudo-Labeling Module (PLM).** We believe that negative proposals, which have a low overlap with the ground truth boxes, may contain objects of potential novel classes. Therefore, we introduce the Pseudo-Labeling Module (PLM) to avoid treating these proposals as background. As shown in Fig. 2(b), we first extract the CLIP representations (named as region embeddings in the figure) of the negative proposals using the approach depicted in Fig. 2(c). Then, we calculate the cosine similarity scores between the region embeddings and the text embeddings of the object class names. After obtaining the similarity scores, we filter out those proposals classified as base classes or background classes and those with CLIP scores lower than a threshold. The remaining few proposals can be identified as novel objects, and we randomly select K proposals as pseudo labels and change the classification target of these proposals during training. We typically set K to 4 by the trade-off of computation and final results. The targets of the positive proposals remain unchanged in this process. More details can be found in the appendix.

**Deployment on RPN and RoIHead.** During the RPN stage, it typically produces numerous negative proposals due to the dense anchor head. To reduce the computation cost of the PLM, we leverage Non-Maximum Suppression (NMS) to eliminate redundant proposals and limit the number

of negative proposals to a fixed amount, such as 1000. In contrast, there are fewer negative proposals during the RoIHead stage than in the RPN stage. Therefore, we send all negative proposals to the PLM. The PLM will change the target of some negative proposals. We convert the classification target from background to foreground for the RPN head, a class-agnostic region proposal network. For the classification branch of RoIHead, we change the classification target of negative proposals from background to pseudo labels generated by the PLM. Note that we only apply the pseudo labels produced by PLM to classification losses. The box regression and mask losses are unchanged.

**Offline Refinement For Self-Training.** Moreover, we propose an offline self-training strategy for OVOD in addition to dynamic self-training. This strategy involves using the trained model to predict the training dataset. The prediction for novel classes with high classification scores will be saved as pseudo labels. The pseudo labels with origin base annotations in the training set will be served as new training sets. The offline refinement works like the teacher-student model. It is only applied after the detector is trained. The generated high-quality pseudo labels are dumped as new supervision. In Sec. 4.3, we present evidence demonstrating the effectiveness of both approaches. Both PLMs and offline refinement serve as two components of our DST framework.

**Discussion.** The proposed PLM is only used during the training without introducing extra parameters and computation costs during the inference. The pseudo-labeling process requires a vocabulary that contains potential novel object classes. We can either obtain the novel classes from the detection datasets following Detic (Zhou et al., 2022b) or collect a wider range of object classes from external sources, e.g., the classes defined in image classification datasets (Deng et al., 2009). We validate the effectiveness of both choices in Table 2f.

### 3.3 TRAINING AND INFERENCE

The training losses adhere to the default settings of Mask R-CNN (He et al., 2017), which comprises three primary losses: classification loss, box regression loss, and mask segmentation loss. Specifically, only the classification loss will be changed based on the pseudo labels, while other losses will remain unmodified. It is important to note that the generated labels are utilized for recognition purposes rather than localization. Therefore, the final loss is expressed as: $L = \lambda_{\mathrm{ps\_cls}} L_{\mathrm{pl\_cls}} + \lambda_{\mathrm{box}} L_{\mathrm{box}} + \lambda_{\mathrm{mask}} L_{\mathrm{mask}}$. $\lambda$ denotes the weight assigned to each loss, and $L_{\mathrm{pl\_cls}}$ is our modified pseudo label classification loss. Specifically, we set a weight of 0.9 for the background class and 1.0 for all other classes. For inference, we adopt the same score fusion pipeline (Equ. 4) as stated in Sec. 3.1.

## 4 EXPERIMENT

### 4.1 EXPERIMENT SETTINGS

**Datasets and Metrics.** We carry out experiments on three detection datasets, including OV-LVIS (Gu et al., 2022), OV-COCO (Zareian et al., 2021), and recent challenging OV-V3Det (Wang et al., 2023a). For LVIS, we adopt the settings proposed by ViLD (Gu et al., 2022) and mainly report the $AP_r$, representing the AP specifically for rare categories. We report box AP at a threshold of 0.5 for COCO and mean AP at threshold $0.5 \sim 0.95$ for V3Det.

**Implementation Details.** We follow the settings of F-VLM (Kuo et al., 2023) for a fair comparison. We use the Mask R-CNN (He et al., 2017) framework with feature pyramid network (Lin et al., 2017) as our detector. All the class names are transferred into CLIP text embedding, following (Gu et al., 2022). For the "background" category, we input the word "background" into the multiple templates prompt and get a fixed text embedding from the CLIP text encoder. We only use the base boxes for training and all class names as of the known vocabulary as the pre-knowledge of PLM. For the final results on LVIS, we train the model for 59 epochs. For ablation studies, we train the model for 14.7 or 7.4 epochs. For the COCO dataset, we follow the previous works (Kuo et al., 2023). For V3Det dataset, we adopt the default setting of the origin baselines (Wang et al., 2023a). More details of other dataset training and testing can be found in the appendix.

Table 1: Results on open-vocabulary object detection benchmarks. We achieve state-of-the-art results on both OV-LVIS and V3Det benchmarks.

(a) OV-LVIS benchmark

| Method | Backbone | $mAP_r$ |
|---|---|---|
| ViLD (Gu et al., 2022) | RN50 | 16.6 |
| OV-DETR (Zang et al., 2022) | RN50 | 17.4 |
| DetPro (Du et al., 2022) | RN50 | 19.8 |
| OC-OVD (Rasheed et al., 2022) | RN50 | 21.1 |
| OADP (Wang et al., 2023b) | RN50 | 21.7 |
| RegionCLIP (Zhong et al., 2022) | RN50x4 | 22.0 |
| CORA (Wu et al., 2023b) | RN50x4 | 22.2 |
| BARON (Wu et al., 2023a) | RN50 | 22.6 |
| VLDet (Lin et al., 2023) | SwinB | 26.3 |
| EdaDet (Shi & Yang, 2023) | RN50 | 23.7 |
| MultiModal (Kaul et al., 2023b) | RN50 | 27.0 |
| CORA+ (Wu et al., 2023b) | RN50x4 | 28.1 |
| F-VLM (Kuo et al., 2023) | RN50x64 | 32.8 |
| Detic (Zhou et al., 2022b) | SwinB | 33.8 |
| RO-ViT (Kim et al., 2023b) | ViT-H/16 | 34.1 |
| baseline | RN50x16 | 26.3 |
| DST-Det | RN50x16 | 28.4 |
| baseline | RN50x64 | 32.0 |
| DST-Det | RN50x64 | 34.5 |

(b) OV-COCO benchmark

| Method | Backbone | $AP_{50}^{novel}$ |
|---|---|---|
| OV-RCNN (Zareian et al., 2021) | RN50 | 17.5 |
| RegionCLIP (Zhong et al., 2022) | RN50 | 26.8 |
| ViLD (Gu et al., 2022) | RN50 | 27.6 |
| Detic (Zhou et al., 2022b) | RN50 | 27.8 |
| F-VLM (Kuo et al., 2023) | RN50 | 28.0 |
| OV-DETR (Zang et al., 2022) | RN50 | 29.4 |
| VLDet (Lin et al., 2023) | RN50 | 32.0 |
| RO-ViT (Kim et al., 2023b) | ViT-L/16 | 33.0 |
| RegionCLIP (Zhong et al., 2022) | RN50x4 | 39.3 |
| baseline | RN50x64 | 27.4 |
| DST-Det | RN50x64 | 33.8 |

(c) OV-V3Det benchmark

| Method | Backbone | $AP^{novel}$ |
|---|---|---|
| Detic (Zhou et al., 2022b) | RN50 | 6.7 |
| RegionClip (Zhong et al., 2022) | RN50 | 3.1 |
| baseline | RN50 | 3.9 |
| DST-Det | RN50 | 7.2 |
| baseline | RN50x64 | 7.0 |
| DST-Det | RN50x64 | 13.5 |

## 4.2 MAIN RESULTS

**Results on LVIS OVOD.** Tab. 1 (a) shows the results of our approaches with other state-of-the-art approaches on LVIS dataset. Due to the limited computing power, we use a smaller batch size (64 vs. 256) and shorter training schedule (59 epochs vs. 118 epochs) to build our baseline, which results in 32.0 % mAP, which is lower than F-VLM (Kuo et al., 2023). After applying our methods, we obtain 34.5% mAP on rare classes, about 1.7% mAP higher than F-VLM (Kuo et al., 2023), without introducing any new parameters and cost during inference. Compared with other stronger baselines, including VLDet (Lin et al., 2023) and Detic (Zhou et al., 2022b), our method does not use any extra data and achieves about 0.7-8.2% mAP gains. We also find consistent gains over different VLM baselines.

**Results on COCO OVOD.** Tab. 1 (b) shows the results of our approaches with other state-of-the-art approaches on COCO dataset. We achieve 27.4% novel box AP when adopting the frozen Mask R-CNN baseline. After using our offline refinement for the self-training module, our method can achieve 6.4% improvement on novel box AP. Both experiment results indicate the effectiveness and generalization ability of our approach.

**Results on V3Det OVOD.** V3Det is a more challenging dataset with a larger vocabulary size than both LVIS and COCO. As shown in Tab. 1 (c), our methods achieve new state-of-the-art results on different backbones. In particular, with RN50x64, our method achieves 13.5 % mAP on novel classes, significantly outperforming previous STOA by 6.8 %. Moreover, with different backbones, our methods can still improve the strong baseline via 3.3-6.5%.

## 4.3 ABLATION STUDY AND ANALYSIS

In this section, we conduct detailed ablation studies and analyses on our DST-Det. More results and analysis can be found in the appendix (Sec. 6).

**Effectiveness of PLM.** In Tab. 2a, we first verify the effectiveness of PLM. Adding PLM in RPN obtains 1.9 % mAP improvements, while inserting PLM in RoI heads leads to 1.2% gains. This indicates the novel classes are more sensitive to RPN. As shown in the last row, combining both yields better results, thereby confirming the orthogonal effect of the two classification losses. As a result, the PLM results in a notable improvement of over 3.0%.

Table 2: Ablation studies and comparative analysis on LVIS 1.0 dataset. $PLM^1$ means PLM in RPN and $PLM^2$ means PLM in RoI head. By default, we add two PLMs. We mainly report $AP_r$ for reference. $AP_{all}$ is also used for all classes. All methods use the strong RN50x64 backbone on LVIS. For results on COCO, we report results using RN50.

(a) Effectiveness of PLM.

| baseline | $PLM^1$ | $PLM^2$ | $AP_r$ |
|---|---|---|---|
| ✓ | - | - | 28.3 |
| ✓ | ✓ | - | 30.4 |
| ✓ | - | ✓ | 29.5 |
| ✓ | ✓ | ✓ | 31.8 |

(b) Loss Choices of PLM.

| Setting | $AP_r$ | $AP_{all}$ |
|---|---|---|
| baseline | 28.3 | 30.2 |
| box loss | 10.3 | 13.6 |
| class + box loss | 20.3 | 25.2 |
| class loss | 31.8 | 31.5 |

(c) Training Schedule. e: epoch.

| Setting | Schedule | $AP_r$ |
|---|---|---|
| baseline | 7.4 e | 28.3 |
| w DST | 7.4 e | 32.2 |
| baseline | 14.7 e | 31.2 |
| w DST | 14.7 e | 33.4 |

(d) Effectiveness of Offline Refinement (OR).

| dataset | method | $AP_r$ |
|---|---|---|
| LVIS | baseline + PLM | 31.8 |
| LVIS | w OR | 32.2 |
| COCO | baseline + PLM | 24.5 |
| COCO | w OR | 28.5 |

(e) More Design Choices of PLM.

| CLIP score | K | $AP_r$ |
|---|---|---|
| 0.5 | 4 | 23.3 |
| 0.8 | 4 | 31.8 |
| 0.4 | 4 | 13.2 |
| 0.8 | 15 | 25.6 |

(f) Supervision Vocabulary During Training.

| Source | $AP_r$ | $AR_{all}$ |
|---|---|---|
| base names only | 28.3 | 30.2 |
| using LVIS rare | 31.8 | 31.5 |
| using ImageNet | 31.5 | 31.2 |

**Ablation on Loss Design in PLMs.** In PLM, we only adopt the classification loss of generated labels and ignore the box loss. This is motivated by the fact that most generated proposals are inaccurate, adversely affecting the localization ability developed using normal base annotations. To validate this assertion, we add the box loss into the PLM. As shown in Tab. 2b, we observe a significant drop in both $AP_r$ and $AP_{all}$. Furthermore, our objective is to enable the detector to recognize the novel objects or part of the novel objects. As shown in Fig. 4, repetitive proposals still persist even with filtering operations in PLMs.

**Ablation on Effect of Pseudo Score and Pseudo Label Number.** We adopt CLIP score and training samples K to select the possible proposals, since most proposals are background noises. As shown in Tab. 2e, decreasing the CLIP score and increasing training samples lead to inferior performance. Decreasing the score may involve more irrelevant objects or context. Increasing training examples may result in more repeated and occluded objects. Moreover, we also visualize the embeddings of novel proposals during the training with different K in Fig. 3(a). With more training examples, the novel class embeddings become less discriminative, contributing to increased noise. Thus, we set K to 4 and the CLIP score to 0.8 by default.

**Ablation on Offline Refinement.** In Tab. 2d, we further show the effectiveness of offline refinement (OR). On both COCO and LVIS datasets, we find the improvements. However, the improvement of COCO is more significant than LVIS. This is because the total number of novel objects in COCO is larger than in LVIS. In summary, both OR and PLM work in complementary, which indicates that simple offline refinement is not a trivial extension.

**Ablation on Various Schedules.** In Tab. 2c, we verify the effectiveness of longer schedule training. After extending the schedule into 14.7 epochs, we still observe over 2.0% improvements. Finally, we extend the training schedule to 59 epochs and still obtain 2.5% improvements, as shown in Tab. 1 (a). This indicates our method can be scaled up with longer training and stronger baselines.

**Ablation on Source of Vocabulary.** We also verify the vocabulary source for novel classes in Tab. 2f. We experiment with two types of sources: the classes (including rare classes) defined in LVIS (Gupta et al., 2019) and the classes defined in ImageNet (Deng et al., 2009). As shown in Tab. 2f, our approach achieves considerable performance even when we *do not* have access to the novel classes in the test dataset and use an external source of potential novel classes, i.e., ImageNet.

**Qualitative Novel Class Example in DST.** We further visualize the generated novel examples in Fig. 3. Most visual examples exhibit satisfactory quality after the filtering operation in PLMs. Compared with previous methods, which treat the novel classes as background, our method directly forces the detector to train the classification head with the assistance of VLM. Despite overlapping

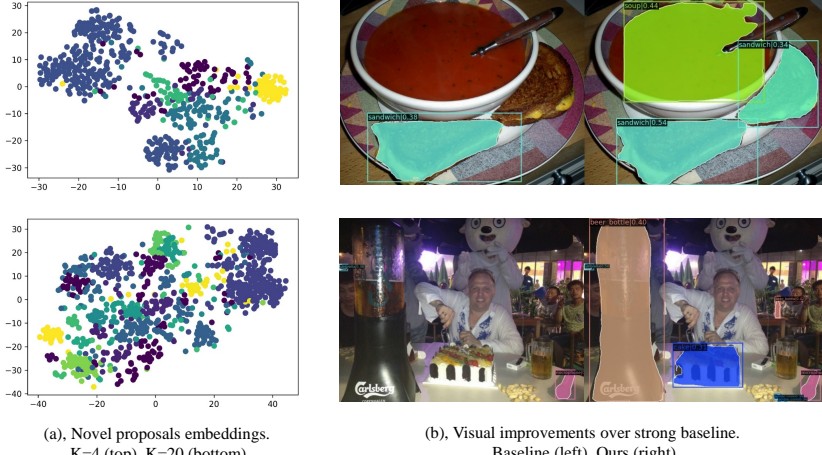

(a), Novel proposals embeddings.
K=4 (top), K=20 (bottom)

(b), Visual improvements over strong baseline.
Baseline (left), Ours (right)

Figure 3: Visual Analysis of DST framework. (a), We present a t-SNE analysis on the novel region embeddings during training. Different color represents different classes. We find that using fewer training samples works well. (b), We show visual improvements over the strong baseline. Our method can detect and segment novel classes, as shown on the right side of the data pair.

or repetitive proposals, we can successfully train the detector by modifying only the classification loss within the RPN and RoI heads. Additional visual examples are presented in the appendix.

**Visual Improvements Analysis.** In Fig. 3 (b), we present several visual improvements over a strong baseline (32.0 $AP_r$) on LVIS. Compared to the baseline, our method demonstrates effective detection and segmentation of novel classes, such as soup and beer bottles.

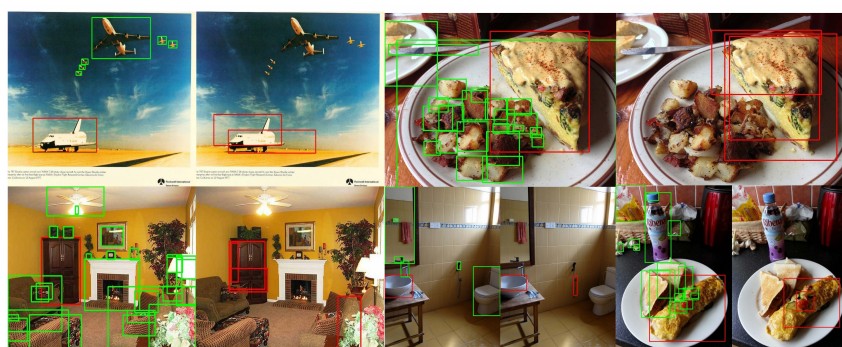

Figure 4: Visual examples. Left: We visualize the class-agnostic ground truth bounding boxes. The green boxes represent the ground truth of base classes and will be used as foreground supervision, while the red boxes represent the ground truth of possible novel classes that are not allowed during training. Right: The red boxes represent the pseudo labels we selected from negative proposals.

## 5 CONCLUSION

This paper presents DST-Det, a novel open vocabulary object detection method that incorporates a dynamic self-training strategy. By rethinking the pipeline of previous two-stage open vocabulary object detection methods, we identified a conceptual gap between training and testing for novel classes. To address this issue, we introduce a dynamic self-training strategy that generates pseudo labels for novel classes and iteratively optimizes the model with the newly discovered pseudo labels. Our experiments demonstrate that this approach can significantly improve the performance of mask average precision (AP) for novel classes, making it a promising approach for real-world applications. We hope our dynamic self-training strategy can help the community mitigate the gap between the training and testing for the OVOD task.

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

## 6    APPENDIX

In this supplementary, we provide the following details:

- Sec. A presents more training, evaluation, and PLM module details.
- Sec. B report more ablation studies and detailed results of three OV-detection benchmarks.
- Sec. C gives more visualization results, including pseudo labels during the training, visual improvements, and failure case examples.

We discuss the limitations and broader impact of our DST-Det at the end of the appendix.

## A    MORE MODEL DETAILS.

**Dataset and Training Details on OV-V3Det.** OV-V3Det (Wang et al., 2023a) is a vast vocabulary visual detection dataset. It contains extremely large categories, which consist of 13,029 categories on real-world images. The train split of V3Det 1,361,181 objects in 184,523 images, the val split has 178,475 objects in 30,000 images, and the test split has 190,144 objects in 30,000 images. For the open vocabulary object detection setting, V3Det randomly samples 6,501 categories as base classes $C_{base}$ and the remaining 6,528 categories as the novel classes $C_{novel}$. For training a detector for V3Det, we use the SGD optimizer with a weight decay of 1e-4, a momentum of 0.9, and a learning rate of 0.02. We train our model for 2x with a batch size of 32.

**Dataset and Training Details on OV-LVIS.** LVIS (Gupta et al., 2019) has a large vocabulary and a long-tailed distribution of object instances. The LVIS dataset contains bounding box and instance labels for 1203 classes across 100k images from the COCO dataset. The classes are categorized into three sets based on their occurrence in training images – rare, common, and frequent. And for open vocabulary object detection, we adopt the settings proposed by ViLD (Gu et al., 2022). In this approach, annotations that belong to common and frequent categories are categorized as base categories. On the other hand, annotations belonging to rare categories are treated as novel categories. We use the SGD optimizer with a learning rate of 0.36 and a weight decay of 1e-4. We train our model for 46.1k iterations with a batch size of 256.

**Dataset and Training Details on OV-COCO.** We follow the setting of ovr-cnn (Zareian et al., 2021) and split COCO2017 into 48 base classes and 17 novel classes. We train our model in this setting for 11.25k interactions with a batch size of 64. Other settings are the same as OV-LVIS.

**Evaluation Protocol on Each Dataset.** We evaluate the model on the standard COCO, LVIS, and V3Det datasets to assess its performance. We report the box average precision at threshold 0.5 of novel classes and mask average precision for rare classes box mean average precision at different threshold (i.e., $0.5 \sim 0.95$) of novel classes for COCO, LVIS, and V3Det respectively.

**Baseline Models in Main Results.** We follow Mask R-CNN (He et al., 2017) pipeline as our detector and use frozen resnet (He et al., 2016) from the CLIP visual encoder and use the text embeddings from the CLIP text encoder as the classifier. The model will obtain two classification scores for each proposal during inference through the detection and VLM branches. The fused score of these two scores serves as the final score for each proposal.

**Algorithm Details of PLM.** In Alg. 1, we present a more detailed algorithm description for our proposed PLM for RPN and RoI head. We will release our code and model for further research.

**Different baseline: OVR-CNN.** For OVR-CNN, we add the *extra* CLIP vision encoder to extract RoI features that are used in PLM (details can be found in the main paper and Alg. 1). Otherwise, the remaining parts, including the learnable backbone and RoI heads, are the same as the OVR-CNN (Zareian et al., 2021).

## B    MORE DETAILED EXPERIMENT RESULTS

**GFLops and Parameter Analysis.** In Tab. 3, we list the GFLops and number of parameters during training and inference. Our method uses a frozen backbone and a learnable detection head and the

---

**Algorithm 1** Pseudo-Labeling Module

---

**Input**: negative proposal set $P_N$, score $S$ of $P_N$, top-level features $F$ from frozen backbone, text embeddings $t$ for all categories, classification label $L_{cls}$

**Output**: pseudo classification label $L'_{cls}$

1: candidates = []
2: **for** $i, p$ **in** enumerate($P_N$) **do**
3:     **if** $S_i > 0$ **then**
4:         $r_i \leftarrow$ RoIAlign($F, p$) /* region embeddings */
5:         $s_i \leftarrow$ MatMul($r_i, t$)
6:         **if** $\max(s_i) > 0.8$ and $\arg\max(s_i) \in$ novel classes **then**
7:             candidates $\leftarrow i$
8:         **end if**
9:     **end if**
10: **end for**
11: selected_idx $\leftarrow$ randomly select K proposals from candidates
12: $L'_{cls} \leftarrow L_{cls}$
13: $L'_{cls}[\text{selected\_idx}] \leftarrow$ foreground or pseudo novel classes
14: Use $L'_{cls}[\text{selected\_idx}]$ to train the RPN and RoI head for the classification branch.

---

Table 3: GFLops and Parameter Analysis. Our method does not bring extra flops or parameters during inference.

|       | Method   | Backbone      | GFLops | Learnable Param. | Frozen Param. |
|-------|----------|---------------|--------|------------------|---------------|
| Train | Baseline | CLIP RN50     | 343G   | 22.9M            | 38.3M         |
|       |          | CLIP RN50x64  | 1565G  | 23.9M            | 420M          |
|       | Ours     | CLIP RN50     | 561G   | 22.9M            | 38.3M         |
|       |          | CLIP RN50x64  | 4961G  | 23.9M            | 420M          |
| Inference | Baseline | CLIP RN50  | 813G   | 22.9M            | 38.3M         |
|       |          | CLIP RN50x64  | 8157G  | 23.9M            | 420M          |
|       | Ours     | CLIP RN50     | 813G   | 22.9M            | 38.3M         |
|       |          | CLIP RN50x64  | 8157G  | 23.9M            | 420M          |

parameters mainly come from the backbone. When using a large backbone, CLIP RN50x64 (Radford et al., 2021), the number of learnable parameters accounts for only one-twentieth of the total number. During training, our method will get 512 proposals per image and obtain its embeddings through RoIAlign (He et al., 2017) and *AttentionPool* operation of CLIP. Compared to the baseline model, our method contributes a significant amount of GFLops from *AttentionPool* operation. During inference, we use the same inference pipeline in the baseline method and use 1000 proposals per image. And a larger number of proposals leads to a higher GFLops.

**More Ablation on Offline Refinement.** In Tab. 4a and Tab. 4b, we conduct a more detailed analysis of our proposed offline refinements. In Tab. 4a, we find that the OR greatly improves COCO dataset performance, while the limited improvement is on large vocabulary datasets. This is because the detectors trained on the COCO dataset better recall novel objects, which is also verified by previous works (Gu et al., 2022). Since our PLM already improves both AR and AP in LVIS and V3Det, the gains of OR are limited. We argue a stronger detector will have more gains. We also explore the score in OR in Tab. 4b, where we find keeping a higher score is important in offline pseudo labels generation.

**Detailed Comparison on LVIS, COCO, and V3Det.** We present more detailed comparison in three OV-benchmarks in Tab. 5, Tab. 6, and Tab. 7. Our method **does not** introduce extra datasets or components compared with other works.

Table 4: More Ablation Studies on Offline Refinement (OR).

(a) Effectiveness of OR on different datasets.

| Method | Backbone | Dataset | $AP_{novel}$ | gains |
|---|---|---|---|---|
| baseline + PLM | RN50 | COCO | 24.5 | - |
| +OR | RN50 | COCO | 28.5 | 4.0 |
| baseline +PLM | RN50x64 | LVIS | 31.8 | - |
| +OR | LVIS | RN50x64 | 32.2 | 0.4 |
| baseline + PLM | RN50 | V3Det | 6.9 | - |
| +OR | V3Det | RN50 | 7.2 | 0.3 |

(b) Effectiveness of score in OR on COCO.

| Setting | $AP_{novel}$ | $AP_{all}$ |
|---|---|---|
| baseline + PLM | 24.5 | 38.3 |
| using GT box | 34.5 | 40.5 |
| score = 0.9 | 28.5 | 38.5 |
| score = 0.7 | 27.2 | 37.8 |
| score = 0.5 | 23.3 | 37.5 |

Table 5: The detailed results of the COCO dataset. We use the 48 and 17 categories from all 80 categories as base and novel classes, respectively, following previous work (Zareian et al., 2021). The Novel AP and AP metrics indicate the average precision (AP50) for 17 novel classes and all 48+17 classes, respectively. Additionally, PLM and OR represent the pseudo-labeling module and offline refine. We only report several representative works in this table.

| Method | Vision Backbone | Trainable Backbone | Extra Data | $AP_{50}^{novel}$ | $AP_{50}^{base}$ | $AP_{50}$ |
|---|---|---|---|---|---|---|
| OVR-CNN (Zareian et al., 2021) | RN50 | ✓ | COCO Captions | 22.8 | 46.0 | 39.9 |
| Detic (Zhou et al., 2022b) | RN50 | ✓ | COCO Captions | 27.8 | 47.1 | 45.0 |
| VL-PLM (Zhao et al., 2022) | RN50 | ✓ | - | 34.4 | 60.2 | 53.5 |
| OV-DETR (Zang et al., 2022) | ViT-B/32 | ✓ | - | 29.4 | 61.0 | 52.7 |
| MEDet (Chen et al., 2022) | ViT-B/32 | ✓ | COCO Captions | 32.6 | 54.0 | 49.4 |
| CORA (Wu et al., 2023b) | RN50x4 | ✓ | - | 41.7 | 44.5 | 43.8 |
| F-VLM (Kuo et al., 2023) | RN50 | ✗ | - | 28.0 | 43.7 | 39.6 |
| RegionCLIP (Zhong et al., 2022) | RN50 | ✓ | COCO Captions (Lin et al., 2014) | 26.8 | 54.8 | 47.5 |
| RegionCLIP (Zhong et al., 2022) | RN50x4 | ✓ | CC3M (Sharma et al., 2018) | 39.3 | 61.6 | 55.7 |
| baseline | RN50x64 | ✗ | - | 27.4 | 43.6 | 39.4 |
| DST-Det (Ours) | RN50x64 | ✗ | - | 33.8 | 56.4 | 50.5 |

## C MORE VISUALIZATION

**Pseudo Labels Visualization.** This section presents additional visualization results. Specifically, we present the pseudo labels generated by our pseudo-labeling module for LVIS (Gupta et al., 2019) and COCO (Lin et al., 2014) in Fig. 5 and Fig. 6. The visualization results are alongside the ground truth annotations for comparison. Our pseudo labels can successfully recall the novel class annotations not seen during training. We also present the visual examples from PLM in the V3Det dataset in Fig. 7.

**Improvement Visualization** In Fig. 8, we show some visualization results predicted by the baseline (left) and our method (right). Our method can accurately detect and segment novel classes with greater precision.

**Failure Cases Visualization.** In Fig. 9, we present several failure examples on LVIS. Although our method has introduced the pseudo-labeling module for novel classes during training, many objects of novel classes cannot be detected or classified correctly.

**Boarder Impacts.** Our method designs the first dynamic self-training on OVOD via mining the self-knowledge from the VLM model and RPN heads. Without extra training data or learnable components, we obtain significant boosts among strong baseline on COCO and LVIS datasets. Our method can be easily extended into other related domains, including open vocabulary instance/semantic segmentation. This will be our further work. Rather than achieving STOA results, our goal is to fully explore the potential of VLM in the detector, which makes our method a generalized approach for various VLM and detectors.

**Limitation.** Our proposed approach leverages a better VLMs model to generate high-quality pseudo labels, where the zero-shot ability of VLMs affects the quality of novel class labels. However, with more strong VLMs in the future (Radford et al., 2021; Li et al., 2023), our method is more feasible for large vocabulary applications.

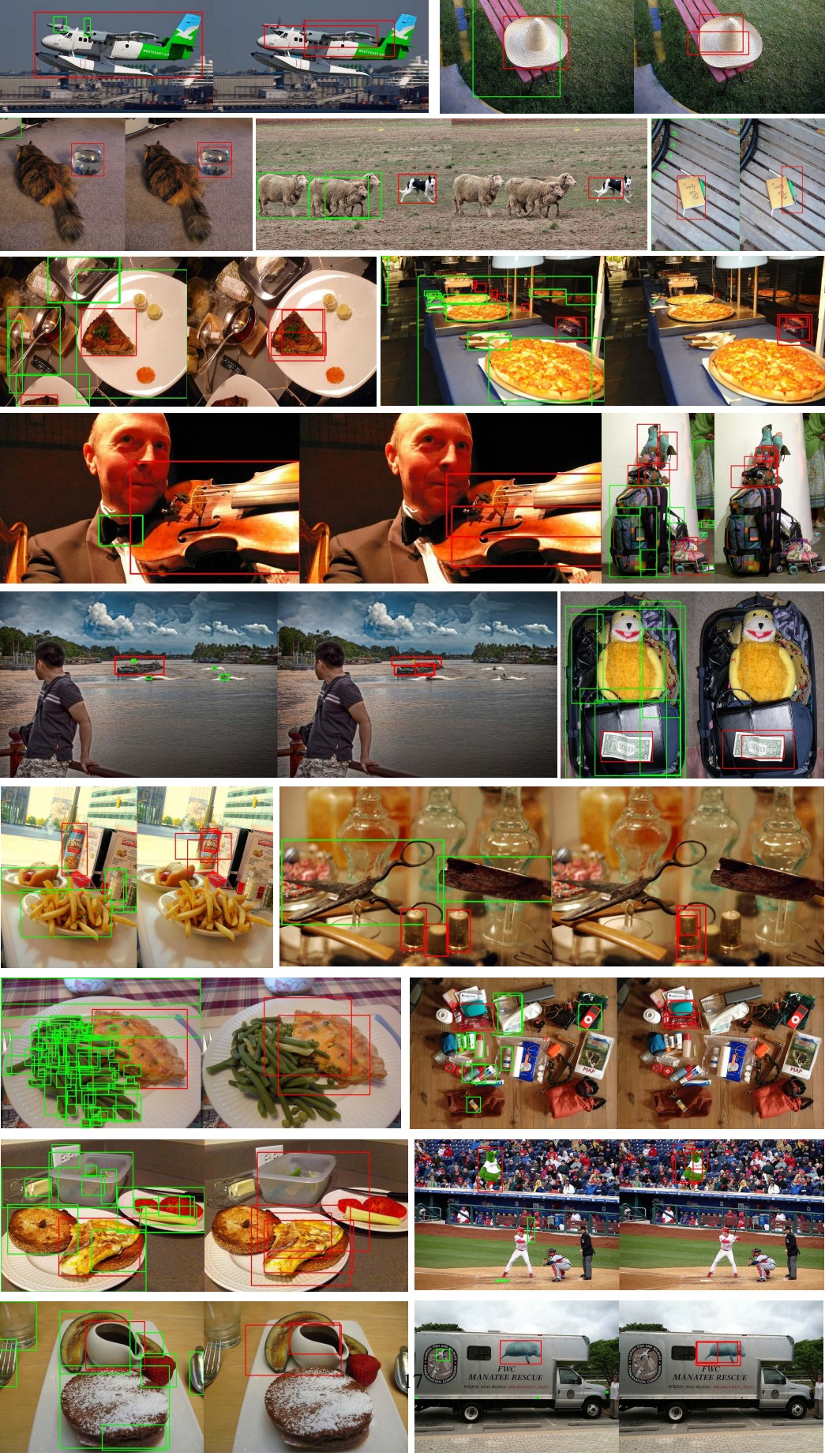

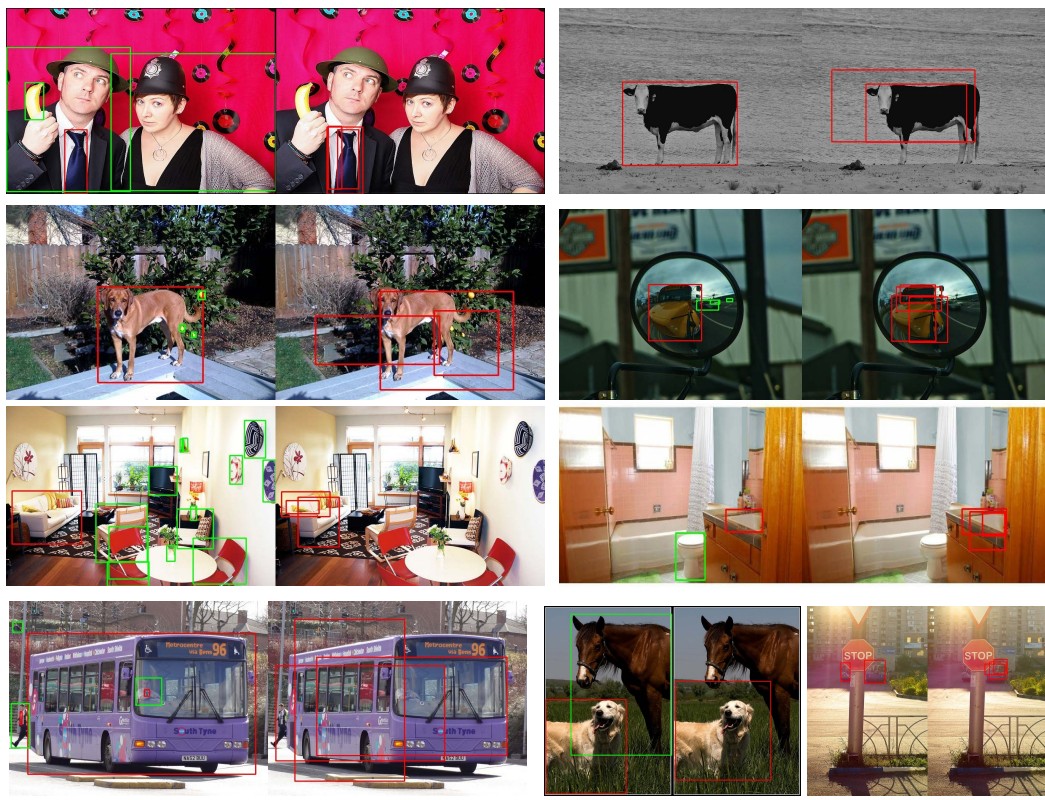

Figure 6: The visualization of pseudo labels on the COCO dataset. The green boxes and red boxes in the left image are the ground truth annotations, and the red boxes in the right image are the pseudo labels.

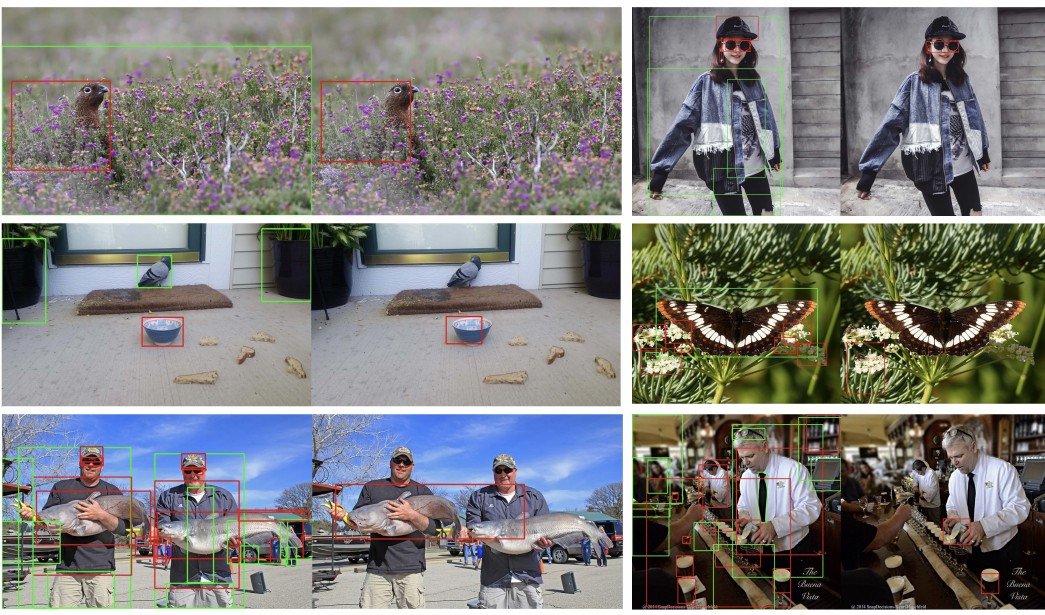

Figure 7: The visualization of pseudo labels on the V3Det dataset. The green boxes and red boxes in the left image are the ground truth annotations, and the red boxes in the right image are the pseudo labels.

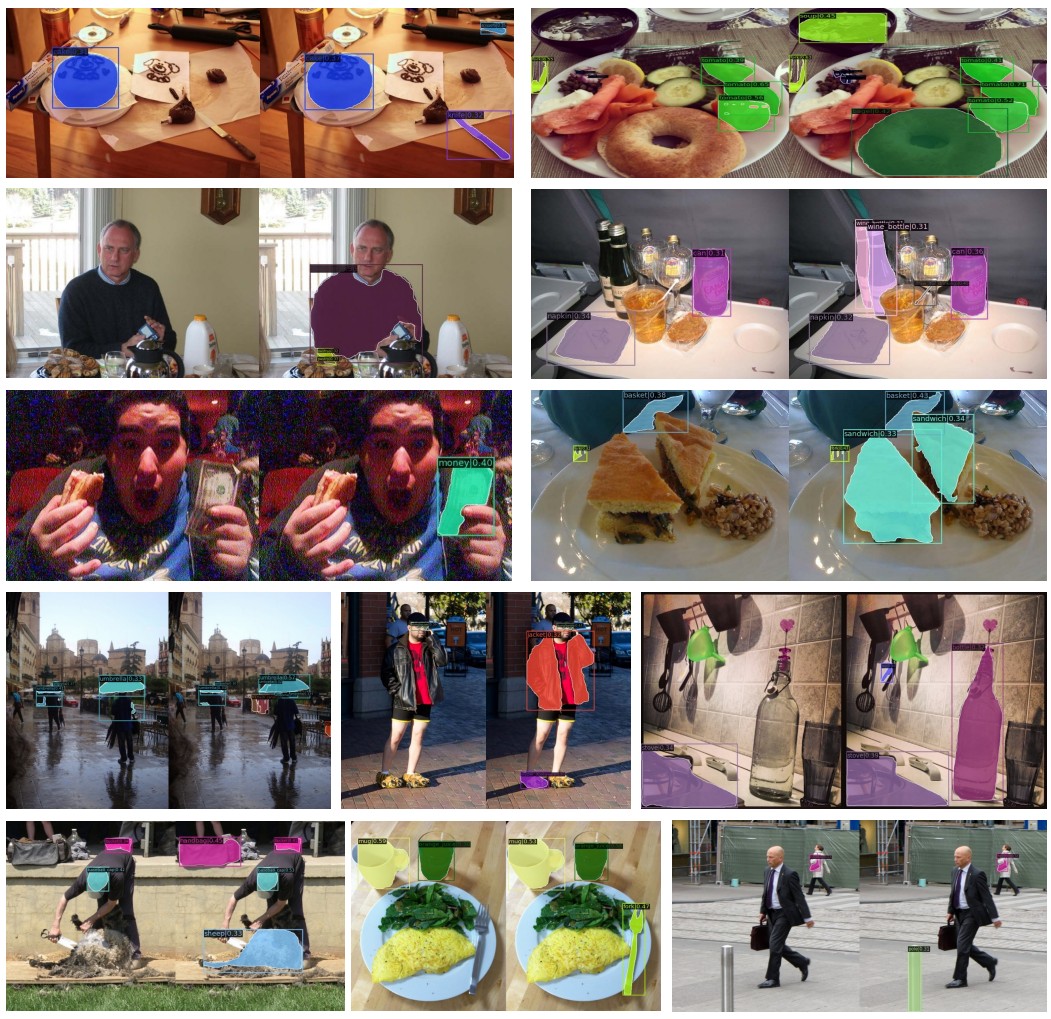

Figure 8: Improvement examples compared to baseline on LVIS dataset. The left is the baseline, and the right is our method.

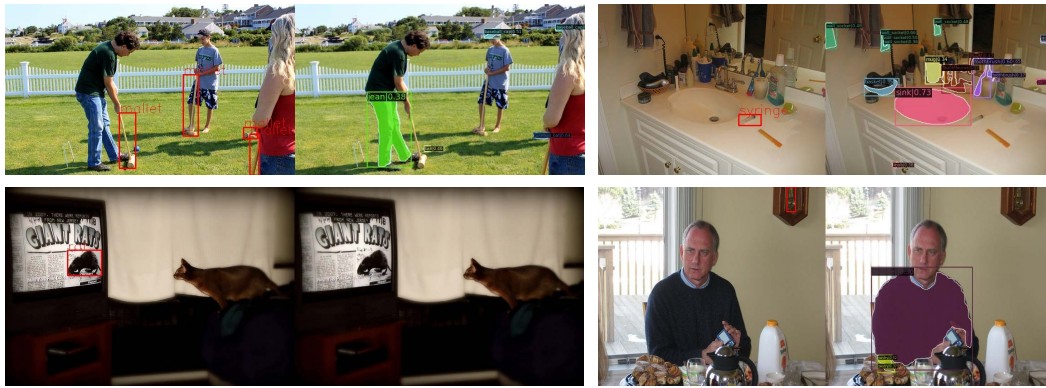

Figure 9: Failure case visualization. The red boxes on the left images represent the ground truth annotations of novel classes; the right images are our predictions.

Table 6: The detailed results of LVIS dataset. We use rare categories as novel classes and frequent and common as base categories. The mask average precision of novel classes is our primary metric. We also report mask AP of all classes for reference.

| Method | Vision Backbone | Trainable Backbone | Extra Data | $AP_r$ | $AP_c$ | $AP_f$ | $AP_{all}$ |
|---|---|---|---|---|---|---|---|
| ViLD-ens (Gu et al., 2022) | RN50 | ✓ | - | 16.6 | 24.6 | 30.3 | 25.5 |
| RegionCLIP (Zhong et al., 2022) | RN50 | ✓ | CC3M (Sharma et al., 2018) | 17.1 | 27.4 | 34.0 | 28.2 |
| Detic (Zhou et al., 2022b) | RN50 | ✓ | IN-L (Deng et al., 2009) | 19.5 | - | - | 30.9 |
| OV-DETR (Zang et al., 2022) | RN50 | ✓ | - | 17.4 | 25.0 | 32.5 | 26.6 |
| DetPro (Du et al., 2022) | RN50 | ✓ | - | 19.8 | 25.6 | 28.9 | 25. |
| F-VLM (Kuo et al., 2023) | RN50 | ✗ | - | 18.6 | - | - | 24.2 |
| BARON (Wu et al., 2023a) | RN50 | ✓ | - | 22.6 | 27.6 | 29.8 | 27.6 |
| OADP (Wang et al., 2023b) | RN50 | ✓ | - | 21.7 | 26.3 | 29.0 | 26.6 |
| GridCLIP (Lin & Gong, 2023) | RN50 | ✓ | - | 15.0 | 22.7 | 32.5 | 25 |
| Detic (Zhou et al., 2022b) | Swin-B | ✓ | IN-L (Deng et al., 2009) | 23.9 | 40.2 | 42.8 | 38.4 |
| VLDet (Lin et al., 2023) | Swin-B | ✓ | CC3M (Sharma et al., 2018) | 26.3 | 39.4 | 41.9 | 38.1 |
| F-VLM (Kuo et al., 2023) | RN50x64 | ✗ | - | 32.8 | - | - | 34.9 |
| baseline | RN50x64 | ✗ | - | 32.0 | 32.7 | 33.1 | 32.8 |
| DST-Det (Ours) | RN50x64 | ✗ | - | 34.5 | 33.7 | 33.5 | 34.6 |

Table 7: The detailed results of V3Det dataset. We use 6,501 categories as base classes and the remaining 6,528 as novel classes. We concentrate on the average precision of novel classes at the threshold (i.e., $0.5 \sim 0.95$). We re-implement F-VLM using the same codebase for reference.

| Method | Vision Backbone | Trainable Backbone | Extra Data | $mAP^{novel}$ | $mAP^{base}$ | $mAP^{all}$ |
|---|---|---|---|---|---|---|
| Detic (Zhou et al., 2022b) | RN50 | ✓ | IN-L (Deng et al., 2009) | 6.7 | 30.2 | 17.7 |
| RegionClip (Zhong et al., 2022) | RN50 | ✓ | CC3M (Sharma et al., 2018) | 3.1 | 22.1 | 12.6 |
| F-VLM (Kuo et al., 2023) | RN50x64 | ✗ | - | 7.0 | 17.2 | 14.5 |
| DST-Det (Ours) | RN50x64 | ✗ | - | 13.5 | 16.9 | 18.2 |

