# OpenReview forum: "DST-Det: Simple Dynamic Self-Training for Open-Vocabulary Object Detection"
_ICLR.cc/2024/Conference — ICLR 2024 Conference Withdrawn Submission_

### Official Review · Reviewer_ht3G · 2023-10-21

**Soundness:** 2 fair
**Presentation:** 3 good
**Contribution:** 1 poor
**Rating:** 3
**Confidence:** 4

**Summary:**

This paper proposes an OVD training method using frozen CLIP to extract vision and text features as the baseline method F-VLM. Different from the baseline, this work uses the pseudo-class labels from CLIP in the training of RPN and RoIHead, which regards the high-score novel-class proposals as positive samples.

**Strengths:**

In this paper, the author finds that just using the CLIP vision feature can already have a good performance on zero-shot classification results for the novel classes in COCO and LVIS. Thus, pseudo labels are generated from CLIP.

**Weaknesses:**

The main weakness is that compared with the baseline, the only contribution seems to be introducing the pseudo labels from CLIP, which is minor.

**Questions:**

- I’m confused about why are the classification branch and loss still needed in the RoIHead. By matching the text feature and vision feature from CLIP, the class can be obtained, so the classification branch and modified pseudo-label classification loss seem not necessary.
- All the experiments are done with ResNet architecture. Since ViT generates advanced performance in CLIP, why not try with the ViT backbone?

---

### Official Review · Reviewer_Fu38 · 2023-10-30

**Soundness:** 3 good
**Presentation:** 3 good
**Contribution:** 1 poor
**Rating:** 3
**Confidence:** 5

**Summary:**

This paper addresses the problem of Open-vocabulary object detection (OVOD), where a set of base classes are given and the detector has to recognize novel objects that do not appear during the training. A limitation of previous works is that they ignore novel classes during training and treat these objects as background. In the paper, the authors propose DST-Det, a dynamic self-training framework for OVOD, where a Pseudo-Labeling Module (PLM) re-evaluate negative proposals in RPN and RoIHead as positive pseudo-labels if their CLIP scores are higher than a threshold. Moreover, they propose an Offline Refinement self-training strategy which predicts the training data and treats high score predictions as pseudo labels for the self-training stage.

**Strengths:**

1. The motivation is clear and the paper is well-written
2. The method is simple and achieves promising results

**Weaknesses:**

1. The motivation and method appear to be similar to VL-PLM (Zhao et al., 2022), which is not rigorously discussed and compared with in the experiment of the main paper.
2. The proposed method seems to lack novelty. The baseline is from the F-VLM paper, and the PLM module and OR are minor. The whole novelty is in the technique of mining more proposals which can be positive candidates as proposed in VL-PLM
3. The results are not convincing enough, the authors should conduct more experiments to show the effectiveness of the method (details in the questions section).
4. We have to know the novel classes in training to "mine" the proposals. Even you can train in advance 1000 classes of ImageNet which cannot cover all classes. That is somehow not the spirit of open-vocabulary object detection when it can detect any classes in test time only.

**Questions:**

1. The baseline results of the OV-COCO benchmark are far from the F-VLM results (27.4 AP novel with RN50x64 backbone vs 28.0 AP novel with RN50 backbone).
2. The transfer detection experiment would make the paper more comprehensive such as in Object365 (as in ViLD paper).

---

### Official Review · Reviewer_ULPC · 2023-10-30

**Soundness:** 2 fair
**Presentation:** 3 good
**Contribution:** 2 fair
**Rating:** 5
**Confidence:** 3

**Summary:**

This paper focus on Open-vocabulary object detection (OVOD). Different previous works that regard the novel classes as background, this paper generates the pseudo labels for the novel classes. To this end, authors presents a simple yet effective strategy to use the clip feature to assign the class label for the detected proposals, and regard them as the foreground objects during training. There is no additional inference cost. The experiments show the effectiveness on LVIS and V3Det.

**Strengths:**

1.	The motivation of this paper is straightforward, the designed method follows the proposed motivation.
2.	The method is very simple yet efficient, which brings no cost for inference.
3.	The paper is well-written and easy following.

**Weaknesses:**

1.	The experiments presentation is no clear, especially in Table 1. More specifically, what is the detailed architecture for baseline, does it resent with mask rcnn? How about the detectors of other SOTAs. If the authors do not present these in detail, the readers cannot have a fair comparison easily.
2.	Why the method can not achieve the SOTA on CoCo dataset? There is no obvious explaination.
3.	Does all models in Table 1 using offline refinement? The authors only say ‘After using our offline refinement for the self-training module’ in Table 1(b), so what is the training details for the others.
4.	In Section 4.2 Results on LVIS OVOD, authors claim ' We also find consistent gains over different VLM baselines’. Which are the different VLM baselines”. I can only see a word baseline in the table, no specific VLM.
5.	The idea that using CLIP to generate the pseudo labels for proposals is very common.

**Questions:**

See weakness, this paper lacks many experiment details, resulting in some confuse.

---

### Official Review · Reviewer_jftV · 2023-11-01

**Soundness:** 3 good
**Presentation:** 3 good
**Contribution:** 3 good
**Rating:** 5
**Confidence:** 5

**Summary:**

This paper proposes a simple self-training approach that uses pretrained VLM (e.g. CLIP) to discover object proposals of possible novel classes. Unlike previous approaches that ignore novel objects during detection training, this approach selects proposals based on specific design criteria. The approach does not need additional annotations or datasets.

**Strengths:**

This paper proposes a simple self-training strategy that does not require additional dataset or multi-stage training. The model leverages the pretrained VLM backbone to discover novel objects which are used to train the detectors on the fly. This idea is intuitive and shows reasonable gains on the baseline according to Table 1 a), b), and c). Table 2 a) presents a nice ablation on LVIS to demonstrate the effectiveness of PLM on both RPN and ROI heads, and adding both yields the best performance.

**Weaknesses:**

The PLM method section briefly describes the proposal filtering mechanism e.g. "filtering out those proposals classified as base classes or background classes or those with low CLIP scores. The remaining few proposals can be identified as novel objects ..." Can the authors expand more on how those proposals are classified as base/background classes? Does "low CLIP scores" mean max(CLIP scores) over classes less than some threshold? (Algorithm 1 in Appendix B seems to suggest that). I think it'd be good to clarify the method in more details as this is core to the contribution of the paper.

In Algorithm 1 of Appendix B, we see the use of novel class names in training phase. It is not clear from the method section whether this is the default setting of the paper. It'd be great to make the use of vocabulary more clear in method and figure 2.

**Questions:**

1. Table 2.f shows that using some novel class vocabulary is very beneficial +3 APr compared to the base names only. Could the authors clarify what vocabulary is used for reporting the result in Table 1 a, b, c? For Table 1a results, the gap between baseline and DST-Det is +2.1 and +2.5 for R50x16 and R50x64, both of which are smaller than the gap of using vocabulary in Table 2.f. Is the benefit of overall system smaller than changing the vocabulary alone?

2. It's interesting that OR is necessary for COCO (see Table 2.d) but not for LVIS. Can the authors explain more why the number of novel objects make a clear difference here? If one enable the PLM loss later in the training schedule (so that the model is somewhat converged), would that make PLM and OR more equivalent? In my view, it'd be a cleaner system/story if OR proves not necessary on top of PLM. In the related work, the authors mention that the method does not require extra VLM tuning and large caption datasets. It seems that the use of OR makes the proposed method more complicated and similar to existing multi-stage pseudo-labeling techniques.

3. What's the variance of the proposed method on LVIS open-vocabulary benchmark over e.g. 3 or 5 independent runs?

---

### Official Review · Reviewer_Sap3 · 2023-11-06

**Soundness:** 3 good
**Presentation:** 2 fair
**Contribution:** 2 fair
**Rating:** 3
**Confidence:** 4

**Summary:**

This paper proposes to use CLIP to identify potential novel class objects during the open-vcabulary object detector training process, which  helps improve the recall and accuracy of novel classes. The paper conducts experiments on several benchmark datasets such as LVIS and demonstrates improvements over baselines.

**Strengths:**

The motivation is clear, and the experiments is conducted with several benchmarks including the new V3det. The paper provides a lot of qualitative results.

**Weaknesses:**

1. The contribution of this paper is weak, only identify that the CLIP cues could help identify more novel classes than the class-agnostic RPN detector.

2. The organization and presentation is weak, which further weakens its contribution and novelty. The major diagrams and captions are not straightforward to help understand the major contribution. Instead appears to be vague and trivial.

3. The experiment results is not very competitive to the SOTA methods, mostly have minor improvements. And the presentation of main result Table 1 is not clear, with different backbones compared to the prior methods. The baseline is beter moved to the ablation study, and only show the proposed method.

**Questions:**

is it posible to evaluate how many novel objects can be idenfied by the proposed method in the open-vocabulary detection training phase ?

**Details Of Ethics Concerns:**

N.A.